# Coordinates Are NOT Lonely - Codebook Prior Helps Implicit Neural 3D Representations

**Fukun Yin**[1]*
fkyin21@m.fudan.edu.cn

**Wen Liu**[2]*
liuwen.steven.1994@gmail.com

**Zilong Huang**[2]
zilong.huang2020@gmail.com

**Pei Cheng**[2]
peicheng@tencent.com

**Tao Chen**[1]†
eetchen@fudan.edu.cn

**Gang YU**[2]
iskicy@gmail.com

[1]**School of Information Science and Technology, Fudan University, China**

[2]**Tencent PCG, China**

## Abstract

Implicit neural 3D representation has achieved impressive results in surface or scene reconstruction and novel view synthesis, which typically uses the coordinate-based multi-layer perceptrons (MLPs) to learn a continuous scene representation. However, existing approaches, such as Neural Radiance Field (NeRF) [15], and its variants [16, 26, 29], usually require dense input views (i.e. 50-150) to obtain decent results. To relive the over-dependence on massive calibrated images and enrich the coordinate-based feature representation, we explore injecting the prior information into the coordinate-based network and introduce a novel coordinate-based model, **CoCo-INR**, for implicit neural 3D representation. The cores of our method are two attention modules: codebook attention and coordinate attention. The former extracts the useful prototypes containing rich geometry and appearance information from the prior codebook, and the latter propagates such prior information into each coordinate and enriches its feature representation for a scene or object surface. With the help of the prior information, our method can render 3D views with more photo-realistic appearance and geometries than the current methods using fewer calibrated images available. Experiments on various scene reconstruction datasets, including DTU [9] and BlendedMVS [28], and the full 3D head reconstruction dataset, H3DS [19], demonstrate the robustness under fewer input views and fine detail-preserving capability of our proposed method.

## 1 Introduction

3D scene reconstruction aims to recover the underlying scene geometry and appearance from a set of posed images. It is a fundamental problem in computer vision and graphics with wide usage in 3D object surface reconstruction and scene novel view synthesis. Recently, implicit neural representation (INR) with coordinated-based learning has been driving significant advantages in producing high-fidelity 3D shapes [16, 26, 29] and photo-realistic images [15, 14, 2]. With the per-scene optimized multi-layer perceptrons (MLPs), such methods usually require a dense set of calibrated views to

---

*Joint first authors.

†Corresponding author.

This work was done when Fukun Yin was an intern at Tencent PCG.

36th Conference on Neural Information Processing Systems (NeurIPS 2022).

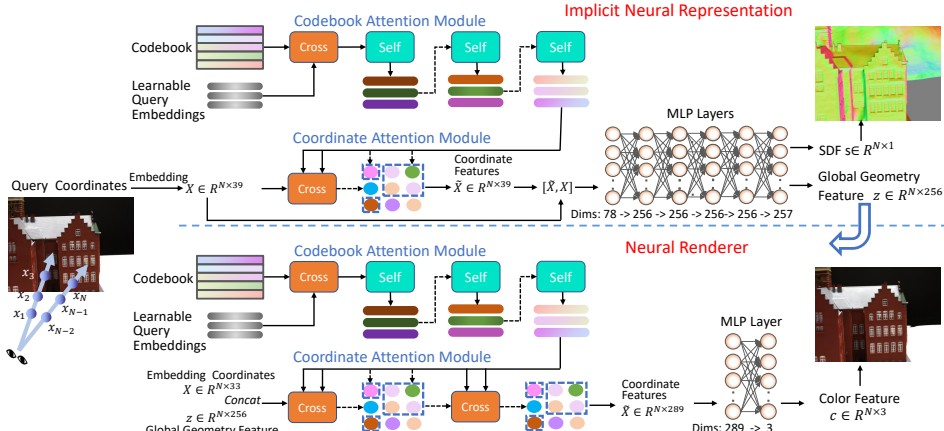

Figure 1: The architecture of the previous and our proposed coordinates network. Our cores are two attention modules: codebook attention and coordinate attention. The former extracts the practical prototypes from the initial codebook, and the latter injects the prototypes into each coordinate.

optimize a high-quality continuous 3D scene representation. However, their performance would degrade dramatically when the number of input views is very limited, because only spatial coordinate information is insufficient for the network to characterize a continuous 3D scene with few input views. It is practical to overcome this drawback by introducing additional high-level semantic descriptors of the context scene as the prior information, to enrich the feature representation of each coordinate and compensate for the missing views when only sparse views are provided.

Fortunately, learning the prior descriptors or prototypes from a large scale dataset is a long-studied topic from the early Bag-of-Words (BoW) in image classification, scene recognition [4] and visual SLAM [11], Vector Quantized (VQ) codebook in data compression [18], to the recent neural discrete codebook representation in image synthesis [23, 20, 3]. These methods learn and maintain a representative codebook, with each codeword denoting a specific prototype from the dataset for serving as a prior to encode new images or scenes. However, how to leverage and correlate the codebook prior with the implicit neural representation is still an unreached area.

This paper investigates a novel coordinate-based deep model that injects the prior codebook information into the implicit neural 3D representation. To achieve this, we first design a **codebook attention** module to extract the valuable prototypes from the prior codebook for each scene. These prototypes might contain rich geometry or appearance information, with parts of them being shared and others being exclusive for different views. The shared prototypes in the codebook guarantee the network to generate view-consistency results. In contrast, the exclusive prototypes for each view bring additional information for the network to enrich its feature representation and produce photo-realistic rendering results and decent geometry by leveraging fewer posed images. Then, we equip each coordinate with the extracted prototypes to improve its feature representation via a **coordinate attention** module. They are both the attention-based modules, as depicted in Fig. 1. The former queries the per-scene relevant prototypes from the codebook prior via the learnable embeddings, and the latter enables each coordinate to query representative features from the prototypes. With the help of prototypes, each coordinate is not alone, resulting in a more robust representation and improving the quality of rendering images.

We perform extensive experiments on different scene reconstruction benchmarks, including DTU [9], BlendedMVS [28] and the full head reconstruction benchmark H3DS [19]. Our approach outperforms current state-of-the-art methods [15, 16, 29] under a sparse set of input views. We summarize the contributions as follows: 1) we propose a novel coordinate-based network that incorporates codebook prior for implicit neural representations, which builds a connection between the coordinates and the codebook prior information for the first time; 2) with the help of the codebook prior, our method could improve the feature representation of each coordinate and is more robust in producing realistic-looking novel views with fewer images; 3) extensive experiments demonstrate the effectiveness of our CoCo-INR, including the robustness in generalizing to novel views under few inputs images and

the capability to produce more photo-realistic results under sparse input views. The code is available at `https://github.com/fukunyin/CoCo-NeRF`

## 2 Related Work and Background

**Codebook Representation**. Bag-of-Words (BoW) is a popular codebook representation method, which has been widely used in document recognition [13] and image classification [22]. A BoW model usually contains the following steps: a) extracting the local features; b) learning the language or visual vocabulary; c) quantizing features by the vocabulary; d) representing a document or an image by the frequencies of words. Recently, Vector Quantization (VQ) has driven advantages in feature representation learning, ranging from the image tokenizer in transformer architecture [25] to the latent codes in generative models [23, 20, 3]. VQ-VAE [23] proposes a simple-but-efficient framework to train an image-level VQ model from a large-scale dataset. VQGAN [3] further improves the codebook representations by using more photo-realistic losses, such as the perceptual loss [10, 31] and the generative adversarial discriminator [5]. Though the VQ codebook provides a powerful tool for generative models, whether it can be explored for implicit neural representation is still unknown.

**Neural Scene Representation and Rendering**. Implicit neural 3D function is leading the development of the shape and appearance reconstruction of a scene [15, 16, 26, 29]. It models the scene as two functions via two neural networks: a geometry function $f : \mathbb{R}^3 \rightarrow \mathbb{R}$ maps the spatial location $x \in \mathbb{R}^3$ to its shape value, and a color function $c : \mathbb{R}^3 \times \mathbb{R}^2 \rightarrow \mathbb{R}^3$ maps a coordinate $x \in \mathbb{R}^3$ and a view direction $d \in \mathbb{R}^2$ to its color value. According to the shape representation differences, we can roughly categorize them into **neural surface fields** [16, 26, 29] and **neural radiance fields** [15, 14, 2]. The former define the object surface as the zero-level set of its signed distance function (SDF) [26, 29] or occupancy [16], shown as follows:

$$S = \{x \in \mathbb{R}^3 | f(x) = 0\}. \tag{1}$$

While the latter uses a generic volume density $\sigma$, and the object surface could be any level set of its density function without constraints, leading to noisy shape reconstruction [29].

Both of these representations enable to render an image via the **volume rendering**. For each ray $\mathbf{r} = \mathbf{o} + t\mathbf{d}$, we sample $N$ points $\{\mathbf{x}_i\}$ along the ray with $\{t_i\}$. Here, $\mathbf{o} \in \mathbb{R}^3$ is the location of the camera and $d \in \mathbb{R}^3$ is the viewing direction. The color $\hat{C}$ of each ray can be approximated as follows:

$$\hat{C} = \sum_{i=1}^{N} T_i \alpha_i c_i = \sum_{i=1}^{N} (\prod_{j=1}^{i-1} (1 - \alpha_i)) \alpha_i c_i, \tag{2}$$

where $T_i$ is the accumulated transmittance along the ray, defined as $T_i = \prod_{j=1}^{i-1}(1 - \alpha_i)$.

The way to calculate the $\alpha_i$ varies among different methods. In NeRF [15], $\alpha_i = 1 - \exp(-\sigma_i \delta_i)$, $\sigma_i$ is the density value at $x_i$, and $\delta_i = t_{i+1} - t_i$. In UNISURF [16], $\alpha_i$ is the predicted occupancy probability. In NeuS [26] and VolSDF [29], a $\sigma$-function is utilized to map the SDF to the volume density probability to calculate the $\alpha_i$. The former chooses a logistic density distribution, and the latter uses a cumulative distribution function of the Laplace distribution as the $\sigma$-function, respectively. These methods can produce photo-realistic images and precise shapes with a dense set of calibrated views. However, their performance will degrade quickly when only sparse or few views are available.

## 3 Methodology

Our goal is to design a new coordinate-based network for implicit neural 3D representation, which utilizes the codebook prior information to improve the feature representation of each coordinate. To demonstrate the effectiveness of our proposed Codebook Coordinate Attentional Implicit Neural Representation (CoCo-INR), we apply it to the task of multi-view scene reconstruction and novel view synthesis. Since our method relies on a representative codebook prior, we employ the same setup of VQGAN model [3] and use a powerful pre-trained model on the ImageNet dataset [21], which contains a codebook $\mathcal{E} \in \mathbb{R}^{E \times D}$ with $E$ prototypes, $e_i \in \mathbb{R}^D$. To make a fair comparison with the previous methods for multi-view scene reconstruction and novel view synthesis, we follow the same pipeline with VolSDF [29] including the geometry representation, sampling strategy, and all training losses, but just replace its MLP-based network with our proposed Codebook Coordinate Attentional Implicit Neural Representation (CoCo-INR).

## 3.1 Codebook and Coordinate Attention

We now present the details of our proposed CoCo-INR. As shown in Fig. 1, it contains two attentional modules: codebook attention and coordinate attention. The former queries the per-scene relevant prototypes from the codebook prior via the learnable query embeddings, while the latter injects the prototypes' information into each coordinate and enriches its feature representation.

**Codebook Attention**. Since a global pre-trained codebook might contain irrelevant prototypes for a specific scene, the codebook attention is going to extract the representative prototypes from the codebook for this scene. Given a pre-trained codebook $\mathcal{E} \in \mathbb{R}^{E \times D}$ where $E$ is number of prototype vectors, and $D$ is the dimension of each prototype vector $e_i$, we define $M$ as the learnable query embedding vectors $\{q_1, q_2, ..., q_M\}$, and each embedding vector $q_i \in \mathbb{R}^S$ queries the scene-relevant prototype $z_i^{(0)} \in \mathbb{R}^S$ from the global codebook $\mathcal{E}$ via a cross-attention mechanism. We will arrive at the initial scene-relevant prototypes, $\mathcal{Z}^{(0)} = \{z_1^{(0)}, z_2^{(0)}, ..., z_M^{(0)}\}$. We then apply $L$ self-attention modules on the initial scene-prototypes $\mathcal{Z}^{(0)}$ to improve their feature representations further and obtain the final scene-relevant prototypes $\mathcal{Z} = \{z_1, z_2, ..., z_M\}$, which is inspired by the previous Perceiver [8, 7] transformer architecture. We describe the procedure of our codebook attention module in Algorithm. 1. **Coordinate Attention**. Through the aforementioned codebook attention module,

---

**Algorithm 1** Codebook Attention Module.

---

**Require:** a global pre-trained codebook, $\mathcal{E} = \{e_i \in \mathbb{R}^D | i \in 1, 2, 3..., E\}$, and the learnable query embedding vectors, $\mathcal{Q} = \{q_i \in \mathbb{R}^S | i \in 1, 2, 3..., M\}$.
**Ensure:** $\mathcal{Z} = \{z_i \in \mathbb{R}^N | i \in 1, 2, ..., M\}$, the output prototype vectors;
1: $Q \leftarrow \{f_Q(q_i) \in \mathbb{R}^{d_k} | i \in 1, 2, ..., M\}$ # $f_Q$ is the query linear projection;
2: $K \leftarrow \{f_K(e_i) \in \mathbb{R}^{d_k} | i \in 1, 2, ..., E\}$ # $f_K$ is the key linear projection;
3: $V \leftarrow \{f_V(e_i) \in \mathbb{R}^{d_k} | i \in 1, 2, ..., E\}$ # $f_V$ is the value linear projection;
4: $\mathcal{Z} \leftarrow$ Cross-Attention$(Q, K, V) =$ Softmax$(\frac{QK^T}{\sqrt{d_k}})V$ # the selected prototypes from the global codebook $\mathcal{E}$;
5: $\mathcal{Z}^{(0)} \leftarrow \{f_O(z_i^{(0)}) \in \mathbb{R}^K | i \in 1, 2, ..., M\}$ # $f_O$ is the output linear projection;
6: **for** $l = 1$ to $L$ **do** # perform $L$ Self-Attention layers to improve prototypes' representation;
7:      # $f_q^{(l)}, f_k^{(l)},$ and $f_k^{(l)}$ are the query, key, and value linear projection, respectively.
8:      $\mathcal{Z}^{(l)} \leftarrow$ Self-Attention$(f_q^{(l)}(\mathcal{Z}^{(l-1)}), f_k^{(l)}(\mathcal{Z}^{(l-1)}), f_v^{(l)}(\mathcal{Z}^{(l-1)}))$
9: **end for**
10: **return** $\mathcal{Z} \leftarrow \{z_i^{(L)} \in \mathbb{R}^S | i \in 1, 2, ..., M\}$ # final scene-relevant prototypes.

---

we will arrive at a series of representative scene-relevant prototypes, $\mathcal{Z}^{(L)} = \{z_1^{(L)}, z_2^{(L)}, ..., z_M^{(L)}\}$. The coordinate attention module is going to gradually inject the representative prototype information into each point via $T$ cross-attention modules, and consequently results in a more representative feature vector for each coordinate. The coordinate attention module is the bridge that connects the coordinate and the codebook prior whose details are shown in Algorithm. 2.

## 3.2 Codebook and Coordinate Attention for Implicit Neural 3D Representations

We now present the details of our proposed CoCo-INR for multi-view scene reconstruction and novel view synthesis. We follow the NeRF++ [30] framework, which separately models the foreground and background neural representations. For the foreground, we follow the neural surface field with VolSDF [29] and replace its MLPs-based network with our proposed codebook and coordinate attentional modules, and arrive at our **CoCo-VolSDF**. Similarly, we use a neural radiance field to model the background scene and arrive at our **CoCo-NeRF**.

CoCo-VolSDF contains two CoCo-based networks: 1) a geometry network, $f_\phi(x) = (s, z)$, predicts the SDF of the object $s \in \mathbb{R}^1$ and a global geometry feature $z \in \mathbb{R}^{256}$; 2) an appearance network, $c_\varphi(x, n, d, z) \in \mathbb{R}^3$, conditions on spatial location, surface normal, viewing direction, as well as the global geometry codes, and then predicts the radiance (color). CoCo-NeRF has a similar architecture, except that the geometry network predicts the density value, and the appearance network only conditions the global geometry feature and the viewing direction as the inputs.

---

**Algorithm 2** Coordinate Attention Module.

---

**Require:** a series of scene-relevant prototypes, $\mathcal{Z}^{(L)} = \{z_i \in \mathbb{R}^C | i \in 1, 2, ..., M\}$ and a set of query coordinates (points), $\mathcal{X} = \{x_i \in \mathbb{R}^C | i \in 1, 2, ..., N\}$;
**Ensure:** $\widetilde{\mathcal{X}} = \{\tilde{x}_i \in \mathbb{R}^C | i \in 1, 2, ..., N\}$, the output coordinate features;
 1: $\mathcal{X}^{(0)} \leftarrow \{x_i \in \mathbb{R}^C | i \in 1, 2, ..., N\}$
 2: # perform $T$ Cross-Attention layers to propagate the prototypes information into each coordinate.
 3: **for** $t = 1$ to $T$ **do**
 4:      $Q \leftarrow \{f_q^{(t)}(x_i) \in \mathbb{R}^{d_k} | i \in 1, 2, ..., N\}$ # $f_q^{(t)}$ is the query linear projection;
 5:      $K \leftarrow \{f_k^{(t)}(z_i) \in \mathbb{R}^{d_k} | i \in 1, 2, ..., M\}$ # $f_k^{(t)}$ is the key linear projection layer;
 6:      $V \leftarrow \{f_v^{(t)}(z_i) \in \mathbb{R}^{d_k} | i \in 1, 2, ..., M\}$ # $f_v^{(t)}$ is the value linear projection layer;
 7:      $\mathcal{X}^{(t)} \leftarrow$ Cross-Attention$(Q, K, V) =$ Softmax$(\frac{QK^T}{\sqrt{d_k}})V$ # bring prototypes for coordinates;
 8:      $\mathcal{X}^{(t)} \leftarrow \{f_o^{(t)}(x_i^{(t)}) \in \mathbb{R}^C | i \in 1, 2, ..., N\}$ # $f_o^{(t)}$ is the output linear projection in attention;
 9: **end for**
10: **return** $\widetilde{\mathcal{X}} \leftarrow \{x_i^{(T)} \in \mathbb{R}^C | i \in 1, 2, ..., N\}$ # final output coordinate features.

---

### 3.3 Training Losses

To train our CoCo-INR, we minimize the difference between the rendered colors and the ground truth colors without 3D supervision. Specifically, we optimize our neural networks by randomly sampling a batch of pixels and their corresponding rays in world space, and this results in a triplet set $\mathcal{S} = \{(C_k, o_k, d_k) | k \in 1, 2, ..., K\}$, where $K$ is the total number of sampled pixels/rays, $C_k$ is its pixel color, $o_k$ and $d_k$ are the camera location and the ray direction from the camera origin to the pixel points in the world space. The loss function is defined as

$$L = L_{rgb} + \lambda L_{eik}. \tag{3}$$

Here, $L_{rgb}$ is color rendering loss, which enforces the rendered pixel color $\hat{C}_k$ computed by Equation. 2 to be similar to the ground truth pixel color $C_k$, formulated as follows:

$$L_{rgb} = \frac{1}{K} \sum_{k=1}^{K} \|\hat{C}_k - C_k\|_1. \tag{4}$$

Since the gradient of SDF satisfies the Eikonal equation $\|\nabla f(x)\| = 1$, we apply this Eikonal term on the sampled points ($N$ in total) to regularize the geometry network $f_\phi$ by

$$L_{eik} = \frac{1}{N} \sum_{i=1}^{N} (\|\nabla f_\phi(x_i)\| - 1)^2. \tag{5}$$

### 3.4 Implementation Details

We use the pre-trained VQGAN [3] model on the ImageNet dataset [21] with a codebook $\mathcal{E} \in \mathbb{R}^{16384 \times 256}$. In each CoCo block, the number of learnable query embeddings is 256, and each embedding has a dimension $e_i \in \mathbb{R}^{128}$. We apply one cross-attention block in the codebook attention module, followed by three self-attention blocks. In the coordinate attention module, we perform one cross-attention operation in the geometry network, and perform once or two times cross-attention operations in the appearance network. All attention modules are the transformer [25] style with multi-head attention mechanism (with four heads), Layer Normalization (Pre-Norm) [1], Feed-Forward Network [25] and GELU activation [6]. The dimension of the key/value and Feed-Forward layer is 128 and 256, respectively. The subsequent fully connected hidden layers use Softplus activation with $\beta = 100$ in the geometry network, and we apply the ReLU activation between hidden layers and Sigmoid for the output in the appearance network.

To model the high-frequency details, we use positional encoding [15] on spatial location as a maximum frequency of 6 and that of view direction as 4. We follow the same hierarchical sampling strategies with the error bound and geometric initialization as VolSDF [29]. Adam [12] based Stochastic Gradient Descent method is used for parameter optimization with a fixed learning rate of 0.0005. The number of sampled rays/pixels is 1024 and the $\lambda$ of $L_{eik}$ is 0.1. Our method is built with Pytorch [17] framework. Each scene is trained on a single Nvidia V100 GPU device for around 3-14 hours, depending on the number of input views.

# 4 Experiments

## 4.1 Experimental Setup

**Datasets**. We conduct the experiments on three publicly available and challenging multi-view scene reconstruction datasets: DTU [9], BlendedMVS [28] and H3DS [19]. The DTU and BlendedMVS are the datasets that contain real objects with different materials, appearances, and geometry. Each scene on the DTU dataset has 49 to 64 posed images with the resolution of $1600 \times 1200$, and there are 31 to 144 calibrated images with the resolution of $768 \times 576$ in the BlendedMVS dataset. Following the previous works [29, 16], we perform experiments on 15 challenging scenes of the DTU dataset and 9 scenes of the BlendedMVS dataset. The H3DS dataset contains 23 scenes of high-resolution full-head 3D textured scans with diverse hairstyles and challenging lighting environments. There are 64 calibrated images in $360°$ views.

**Baselines and Evaluation Metrics**. We choose the recent state-of-the-art implicit neural 3D representation methods, including NeRF [15], UNISURF [16], and VolSDF [29] as the baselines. To evaluate the performance of each method in scene representation, we use three common metrics: Peak Signal to Noise Ratio (PSNR), Structural Similarity (SSIM) [27], and LPIPS [31] on novel view synthesis. Higher PSNR and SSIM mean a better performance, while a lower LPIPS means better.

**Settings with Different Numbers of Views**. Since the previous methods like UNISUIRF [16] and VolSDF [29] use all images of each scene to train the model and report the rendering results on the training views, this is unreasonable to measure the generalization performance in the novel views. To better evaluate the performance of each method under a sparse set of input images, in all datasets, we only use half (or less than half) of the provided images (16-32 in total) of each scene as the training set and set the left views as the testing set. Same with UNISUIRF [16] and VolSDF [29], we do not use the object mask inputs and the mask loss. We further conduct the experiments with fewer training views (5-8 in total) to compare the robustness of each method.

## 4.2 Performance Comparison

To comprehensively evaluate the performance, we compare our method with baselines on DTU, BlenedMVS, and H3DS datasets under different numbers of training views.

**Setting with Sparse Views (16-32 in total)**. The quantitative results under this setting are reported in Table 1. Our CoCo-based method outperforms others in terms of PSNR, SSIM, and LPIPS when sparse views are available. Also, we further illustrate the qualitative comparison with other methods in Fig 2, which demonstrates that our method produces a more realistic-looking novel view synthesis and more precise normal results concerning geometry.

It can be seen from Fig 2 that our method shows robustness and detail recovery ability in the case of sparse views. NeRF's [15] estimated volume density could not generate accurate surface reconstruction and novel views with large-scale changes. UNISURF [16] relies on multiple views to learn details and distinguish foreground and background and is sensitive to noise (The H3DS dataset [19] has challenging lighting environments due to the use of flash). Both ours and VolSDF [29] have shown generalization for a novel view. However, DTU failed in the scene where the foreground and background were indistinguishable (the second column in Fig. 2), because each point is independent and cannot obtain semantic information. Our CoCo-based method queries the per-scene relevant prototypes from the codebook prior via the learnable query embeddings, to deal with scenes with similar colors and features. More importantly, propagating prior information into each coordinate will enrich its feature representation for a scene or object surface, and improve the network's ability to model the high-frequency information. Consequently, our method can preserve the details better, such as the holes (the first column), reflections (the fourth column), hair, and expressions (the last two columns) in Fig. 2.

**Setting with Few Views (5-8 in total)**. To further demonstrate the generalization and the robustness of our method under a few input views, we conduct the experiments and make comparisons with VolSDF [29], which has demonstrated generalization in sparse views. As Table. 2 shows, in this setting, our method has significantly improved over the baseline method, VolSDF, with a higher PSNR, SSIM, and LPIPS, respectively. Also, it is apparent from Fig. 3 that our method is more robust to produce better photo-realistic novel views with more detailed geometry normal than the baseline method, which demonstrates that the codebook prior information could indeed be a compensation for

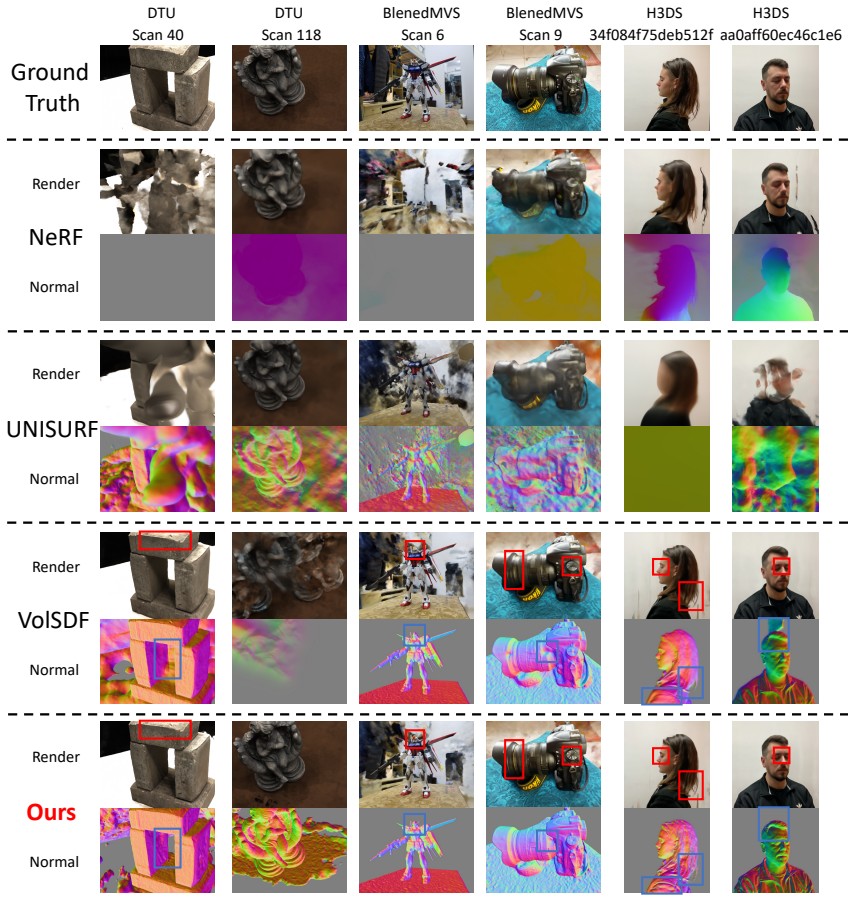

Figure 2: Qualitative results (zoom-in to view better) of each method on the DTU, BlendedMVS, and H3DS datasets under sparse views (16-32 in total). We both visualize the novel view synthesis and the surface normal of each method.

Table 1: The results of different methods on the DTU, BlendedMVS, and H3DS datasets with sparse views (16-32) and without object masks of each scene. ↑ means the higher, the better.

| | DTU | | | BlendedMVS | | | H3DS | | |
|---|---|---|---|---|---|---|---|---|---|
| | PSNR↑ | SSIM↑ | LPIPS↓ | PSNR↑ | SSIM↑ | LPIPS↓ | PSNR↑ | SSIM↑ | LPIPS↓ |
| NeRF [15] | 22.162 | 0.790 | 0.306 | 16.523 | 0.667 | 0.272 | 21.185 | 0.861 | 0.144 |
| UNISURF [16] | 23.549 | 0.823 | 0.348 | 14.749 | 0.649 | 0.299 | 17.095 | 0.816 | 0.190 |
| VolSDF [29] | 26.609 | 0.839 | 0.309 | 18.942 | 0.747 | 0.213 | 23.922 | 0.898 | 0.110 |
| **Ours** | **26.738** | **0.852** | **0.298** | **19.594** | **0.764** | **0.201** | **25.279** | **0.911** | **0.098** |

the limited input views. To investigate the potential reason why our CoCo-based coordinate network could render better realistic-looking views under fewer input views than the MLP-based network, we compare the feature distribution of sampling points.

We take two patches from different areas of the foreground and visualize their geometry features and appearance features by the t-SNE [24] method, see Fig. 4. We can see that our proposed CoCo-based network results in more discriminative features for each coordinate, which indicates that the codebook prior could enrich each coordinate's feature representation. On the contrary, MLP-based networks, such as VolSDF, cannot distinguish the features of different points in both geometric and appearance spaces. In other words, MLP-based networks cannot extract necessary features for surface reconstruction and new views generation, leading to errors in geometry normal or blurring and color distortion.

**Setting with Extremely Limited Views (3 views).** In this part, we attempt a challenge: only use three training views to train our CoCo-based network. In this challenge, three training views mean

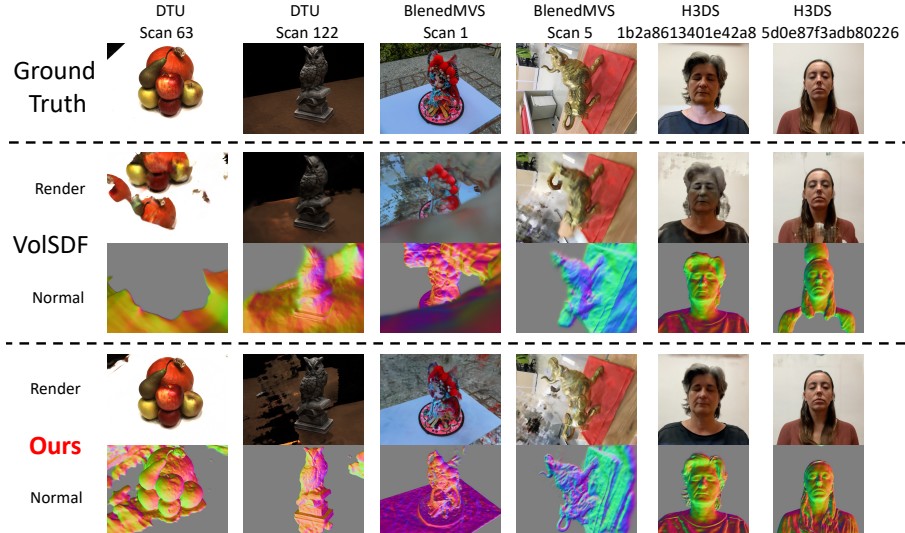

Figure 3: Qualitative visualization results (zoom-in for the best of views) on the DTU, BlendedMVS, and H3DS datasets with few views (5-8). We both show the novel view synthesis and the surface normal of the state-of-the-art method, VolSDF [29] and our method.

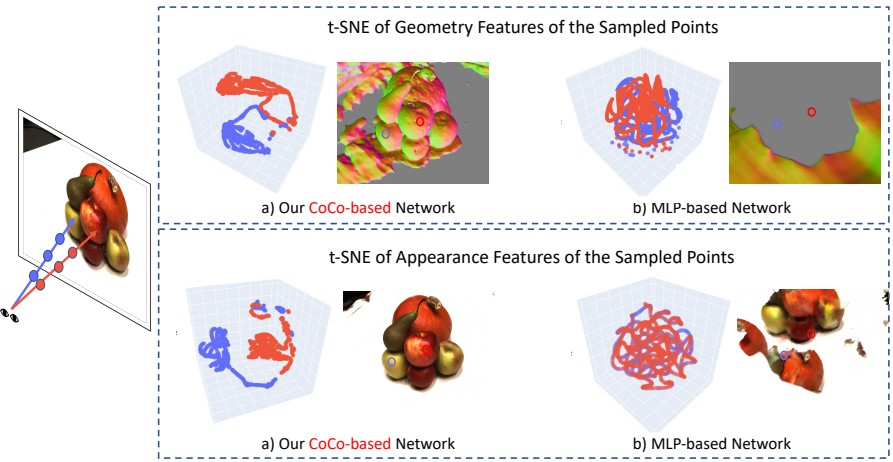

Figure 4: The visualization of learned features of the sampled points, by our CoCo-based network and the MLP-based network under a few input views. Both geometry features and sampled appearance features of all coordinates are illustrated.

that it is difficult for the network to learn background information, so we use masks to make the network more focus on the foreground. We conduct experiments on the DTU dataset and compare with mask-based VolSDF [29]. As shown in Table 3, our method has significantly improved over VolSDF with a higher mean PSNR, SSIM, and LPIPS respectively. Especially for the LPIPS, VolSDF is nearly 25% higher than our method, which means that the new view images generated by our method perform better at the semantic level. We visualize part of the experimental results in Fig. 5, and it can be seen that our CoCo-INR can still render satisfactory RGB images and surface features under the extremely limited 3 training views.

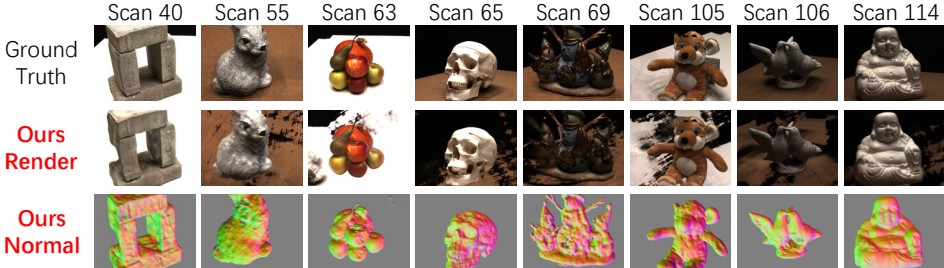

Figure 5: Qualitative visualization results (zoom-in for the best of views) on the DTU dataset with 3 extremely limited views.

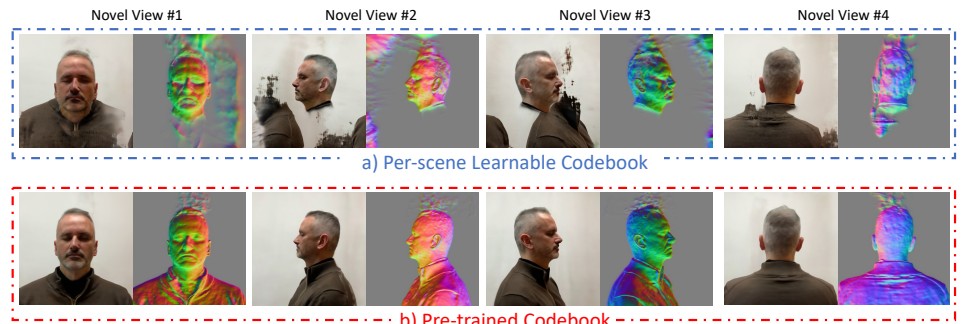

Figure 6: Results of our method with different codebooks with 8 views available on the H3DS dataset.

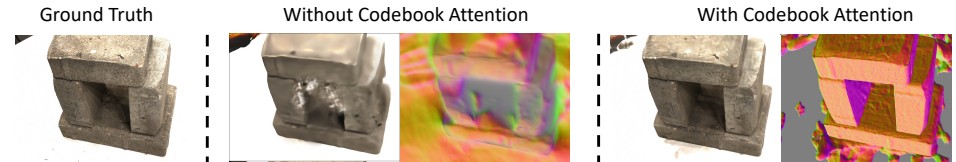

Figure 7: Results of our method with and without codebook attention module on the DTU dataset.

Table 2: The results between VolSDF [29] and ours, on the DTU, BlendedMVS, and H3DS datasets with few views(5-8) and without object masks of each scene. ↑ means the higher, the better.

| | PSNR↑ | SSIM↑ | LPIPS↓ | PSNR↑ | SSIM↑ | LPIPS↓ | PSNR↑ | SSIM↑ | LPIPS↓ |
|---|---|---|---|---|---|---|---|---|---|
| VolSDF [29] | 18.692 | 0.747 | 0.362 | 13.918 | 0.624 | 0.317 | 18.939 | 0.846 | 0.166 |
| **Ours** | **19.624** | **0.757** | **0.346** | **14.417** | **0.630** | **0.298** | **19.084** | **0.847** | **0.165** |

Table 3: The results of different methods on the DTU dataset for each scene with 3 training views.

| Method | PSNR↑ | SSIM↑ | LPIPS↓ |
|---|---|---|---|
| VolSDF [29] | 13.929 | 0.522 | 0.491 |
| **Ours** | **14.761** | **0.624** | **0.396** |

Table 4: Per-scene learnable codebook vs. fixed pre-trained codebook with few views (5-8). Experiments on the first three scans of the DTU dataset(scan24, scan37, and scan40). ↑ means the higher, the better.

| Method | PSNR↑ | SSIM↑ | LPIPS↓ |
|---|---|---|---|
| Per-scene Learnable Codebook | 13.315 | 0.525 | 0.298 |
| **Ours (Fixed Pre-trained Codebook)** | **15.622** | **0.576** | **0.283** |

### 4.3 Ablation Studies and Analysis

**Per-scene Learnable Codebook vs. Fixed Pre-trained Codebook.** To verify the efficacy of the codebook prior, we perform controlled experiments on both a per-scene learnable codebook and a fixed pre-trained codebook from a large-scale dataset, ImageNet [21] by VQGAN model [3]. The comparison result is shown in Table 4 and Fig. 6. Our method with a per-scene learnable codebook does not introduce additional information and would fail under a few input views, which further indicates the importance of a representative codebook in our system.

**Impact of Codebook Attention.** To demonstrate its efficacy, we conduct an ablation study of our CoCo-NIR with and without the codebook attention module. The comparison result in Fig. 7 shows that the codebook attention improves the robustness of our system when the scene has many homogeneous textures, which reflects that the codebook attention could extract scene-relevant prototypes (such as shapes or textures) for the coordinates.

Table 5: Different number of codebook items with few views (5-8). Experiments on the first three scans of the DTU dataset(scan24, scan37, and scan40). ↑ means the higher, the better.

| Method | PSNR↑ | SSIM↑ | LPIPS↓ |
|---|---|---|---|
| 8192 items | 15.463 | 0.567 | 0.287 |
| 4096 items | 14.042 | 0.550 | 0.288 |
| 2048 items | 14.762 | 0.565 | 0.285 |
| 1024 items | 13.948 | 0.548 | 0.300 |
| **Ours(16384 items)** | **15.622** | **0.576** | **0.283** |

**Impact of Codebook Size.** We report the results of different numbers of codebook items, ranging from 16384, 8192, 4096, 2048 to 1024 in Table 5. It shows that the performance tends to decrease with the reduction of the codebook items.

## 5 Conclusions and Limitations

We introduce CoCo-INR, a novel framework for implicit neural 3D representations. It surpasses the previous MLPs-based implicit neural network by building a connection between each coordinate and the prior information, propagates them into each coordinate, and consequently, a more representative feature for each coordinate. With the help of prior information, our framework could produce photo-realistic novel view synthesis and 3D shapes with fewer posed images available in the task of scene reconstruction.

Our framework is flexible and generic. In future work, our framework has the potential ability to handle cross-modalities implicit neural representations such as text-to-image/SDF/NeRF via the connection between the language prompts and coordinate. Moreover, if we replace the per-scene learnable query embeddings in codebook attention modules with the image/pose-dependent feature tokens, our framework could also be extended to a generalizable network across scenes rather than the current per-scene optimization.

## Acknowledgements

This work is supported by National Natural Science Foundation of China (No. 62071127 and 62101137), Zhejiang Lab Project (No. 2021KH0AB05).

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
