# OpenReview forum: "Coordinates Are NOT Lonely - Codebook Prior Helps Implicit Neural 3D representations"
_NeurIPS.cc/2022/Conference — NeurIPS 2022 Accept_

### Official Review · Reviewer_k1QM · 2022-07-03

**Rating:** 6
**Confidence:** 3
**Soundness:** 2 fair
**Presentation:** 2 fair
**Contribution:** 3 good

**Summary:**

The paper proposes the utilization of a pre-trained codebook within an implicit neural 3D representation. This is enabled by using two attention modules termed codebook attention and coordinate attention, which together form the proposed Codebook Coordinate Attentional Implicit Neural Representation (CoCo-INR). The model is evaluated on the DTU, BlendedMVS, and H3DS datasets and compared with NeRF, UNISURF, and VolSDF using either "sparse views" (16-32 views) or "few views" (5-8 views). CoCo-INR shows better performance than the baseline methods in the conducted experiments.

**Questions:**

- What is the intuition behind using the VQGAN pre-trained codebook? These codes are extracted from 2D images, but the paper tackles 3D reconstruction. Would it be better to use a pre-trained codebook from a 3D network?
- There are two coordinate attention modules in the "Neural Renderer" part of the pipeline, does this mean that the codebook is more important for generating the color?
- It would be interesting to include further ablations, for example:
  - reducing the number of MLP layers (maybe even to 0) to see the effect of only using the introduced codebook/coordinate attention modules
  - reduce the number of utilized codes from 16384 to maybe 8192, 4096, and 1024 by random sampling to investigate the effect of the codebook size
  - using a randomly initialized codebook to understand the effect of the VQGAN pre-trained codebook
  - exclude the "skip connection" in the "Implicit Neural Representation" (Suppl., Fig. 1, upper part) that concatenates the output of the coordinate attention module ($\widehat{X} \in R^{N \times 39}$) with the query coordinates after embedding ($X \in R^{N \times 39}$). This would also help to show how only the codebook-based module works.
  - Use more coordinate attention modules in the "Implicit Neural Representation" and fewer in the "Neural Renderer"


**Limitations:**

The limitations are addressed in the supplementary material.

**Strengths And Weaknesses:**

**Strengths**

- The idea of combining a pre-trained codebook with (cross) attention modules is very interesting.
- The paper has high visual quality due to well-arranged figures and tables.
- For the presented experiments the model shows an improvement over the compared-against models (VolSDf, UNISURF, NeRF).

**Weaknesses**

- The comparison wrt. the SOTA is only done on "sparse views" and "few views". While these settings are sensible for evaluation of the proposed approach, it is crucial to also evaluate with pre-existing evaluation protocols. Currently, the reader cannot tell if the improved performance originates from an actually improved model, or because the model and benchmark were designed simultaneously.
-  The codebook idea is very interesting, however, it is difficult to understand what this module adds to the network and where the improved performance originates. Please also see the questions below.
- The ablation study is only qualitative and could contain further ablations, please see the questions below.
- Understanding the proposed architecture could be easier. I found Figure 1. in the supplementary material very helpful because it shows the actual computation the model is doing. In contrast, I found Algorithms 1 & 2 to be less helpful because they mostly reproduce the attention algorithm.

Overall, the paper proposes an interesting idea that seems to improve results. If the evaluation wrt. the SOTA is improved, the ablation study is expanded by more ablations and quantitative results, and the clarity is improved I am willing to upgrade my rating.

---

> ### Author Response · Authors · 2022-08-02
> **Authors Response (Part 3)**
>
> 📝  **Q:  Reduce the number of utilized codes.**
>
> 💡&#8194;**A:**  We report the results of different numbers of codebook items, ranging from 16384, 8192, 4096, 2048 to 1024 in table below. It shows that the performance tends to decrease with the reduction of the codebook items.
>
>   |Method  with few views (5-8)  |PSNR↑  | SSIM↑  | LPIPS↓  |
> |:--:|:--:|:--:|:--:|
>   |8192 items  | 15.463  | 0.567   |0.287  |
>   |4096 items  | 14.042  | 0.550   |0.288  |
>   |2048 items  | 14.762   |0.565   |0.285  |
>   |1024 items  | 13.948  | 0.548   |0.300  |
>   |**Ours(16384 items)**   |**15.622**  | **0.576**   |**0.283**  |
>
> 📝  **Q:  Use a randomly initialized codebook.**
>
> 💡&#8194;**A:** Experiment results of either using randomly initialized codebook or without codebook are shown in table below. The performance and qualitative quality of both settings degrade significantly, and some scans fail with few views. Therefore, our prior can provide stronger robustness and better rendered images, especially with fewer views.
>
>  |Method  with few views (5-8) |PSNR↑  |SSIM↑ | LPIPS↓ |
> |:--:|:--:|:--:|:--:|
>  |Without codebook  |12.694  |0.523  |0.325 |
>  |Randomly initialized codebook  |13.315 | 0.525  |0.298 |
>  |**Ours (pre-trained codebook)**  |**15.622**  |**0.576** | **0.283** |
>
>   |Method with sparse views (16-32)  | PSNR↑  | SSIM↑   |LPIPS↓  |
> |:--:|:--:|:--:|:--:|
>   |Randomly initialized codebook  | 21.458  | 0.615   |0.233  |
>   |**Ours (pre-trained codebook)**   |**22.907**   |**0.688**  | **0.229**  |
>
> 📝  **Q:  Exclude the skip connection.**
>
> 💡&#8194;**A:**  The skip-connection can not be removed in the implicit neural representation because the network requires at least one MLP layer with coordinates information for analytic geometric (spherical) initialization, same with VolSDF.
>
>   |Method with few views (5-8)   |PSNR↑  | SSIM↑  | LPIPS↓  |
> |:--:|:--:|:--:|:--:|
>   |Without skip-connection  | 7.969  | 0.380  | 0.371  |
>   |**Ours (with skip-connection)**   |**15.622**  | **0.576**   |**0.283**  |
>
>   |Method with sparse views (16-32)  |PSNR↑   |SSIM↑  | LPIPS↓  |
> |:--:|:--:|:--:|:--:|
>   |Without skip-connection   |10.088  | 0.454  | 0.369  |
>   |**Ours (with skip-connection)**   |**22.907**   |**0.688**   |**0.229**  |
>
>
> 📝  **Q: Use more coordinate attention modules in the "Implicit Neural Representation" and fewer in the "Neural Renderer"**
>
> 💡&#8194;**A:**  The number of coordinates attention modules in Implicit Neural Representation (geometry) and Neural Renderer (color) networks need to be tuned since there is only the RGB color supervision for the Neural Renderer network. We perform the ablation studies with a different number of modules in table below. We can see that directly using fewer modules in Neural Renderer would decrease the performance due to the reduced color representation. While directly using more modules in Implicit Neural Representation would also reduce the performance since the RGB supervision applied to the color part, instantly increasing the geometry network's parameters might result in overfitting.
>
>   |Method with few views (5-8)  | PSNR↑   |SSIM↑   |LPIPS↓  |
> |:--:|:--:|:--:|:--:|
>   |INR #1, ↓NR #1 |14.840  | 0.560  | 0.291  |
>   |↑INR #2, NR #2  |14.187  | 0.553  | 0.287  |
>   |↑INR #2, ↓NR #1 |13.946  | 0.528  | 0.323  |
>   |**Ours(INR #1, NR #2)**  | **15.622**  | **0.576**  | **0.283**  |

---

> > ### Comment · Reviewer_k1QM · 2022-08-08
> > **Reviewer Response After Rebuttal**
> >
> > I thank the authors for their extensive response and especially for running this many further experiments. The response has addressed my concerns and I will update my rating accordingly. I would encourage the authors to include (if space permits) some of the additional explanations from this response into the main paper because these really helped with my understanding of the paper.

---

> > > ### Author Response · Authors · 2022-08-08
> > > **Authors Response After Rebuttal**
> > >
> > > We appreciate your valuable and insightful comments. We feel glad about your generally favorable assessment of our methodology. Additional evaluation/ablation and corresponding explanations will be included in the final version.

---

> ### Author Response · Authors · 2022-08-02
> **Authors Response (Part 2)**
>
> 📝  **Q:  Does more coordinate attention modules mean that the codebook is more important for generating the color.**
>
> 💡&#8194;**A:**  It is related to the properties of the codebook. Our codebook is trained in 2D and can provide the Neural Renderer network with rich feature representation and texture information, to produce more photo-realistic results. For Implicit Neural Representations, it just requires injecting prior codebook information into each coordinate, resulting in a more robust representation. The former needs to learn textures from rich geometry features and view-consistent prototypes, which is more challenging because human visual perception requires detail-preserving capability.
>
> 📝  **Q:  Reduce the number of MLP layers (maybe even to 0).**
>
> 💡&#8194;**A:**  Experiment results of reducing the number of MLP layers in the Implicit Neural Representations module are shown in table below. Benefiting from our CoCo-Attention module, which queries representative features from per-scene relevant learnable embeddings for each coordinate, new views can be synthesized even with 1 MLP layer under different settings of view sparsity. But using 0 MLP layers will not work because we need at least 1 MLP layer for the spherical geometry initialization as used in VolSDF.
>
>   |Method  with few views (5-8) | PSNR↑   |SSIM↑   |LPIPS↓  |
> |:--:|:--:|:--:|:--:|
>   |2 layers MLP   |14.656   |0.560  | 0.292  |
>   |1 layers MLP  | 13.210  | **0.576**   |0.311  |
>   |0 layers MLP   |7.309  | 0.303   |0.401  |
>   |**Ours (4 layers MLP)** |**15.622**  | **0.576**  | **0.283**  |
>
>   |Method with sparse views (16-32)   |PSNR↑   |SSIM↑   |LPIPS↓  |
> |:--:|:--:|:--:|:--:|
>   |2 layers MLP  | 21.946   |0.592   |0.240  |
>   |1 layers MLP  | 21.171   |0.622  | 0.231  |
>   |**Ours (4 layers MLP)**   |**22.907**  | **0.688**  | **0.229**  |
>
>   |Method with all views  |PSNR↑  | SSIM↑   |LPIPS↓  |
> |:--:|:--:|:--:|:--:|
>   |2 layers MLP  | 26.163  | **0.706**  | 0.227  |
>   |1 layers MLP  | 25.332   |0.644   |0.237  |
>   |**Ours (4 layers MLP)**  | **26.870**   |0.691   |**0.218**  |

---

> ### Author Response · Authors · 2022-08-02
> **Authors Response (Part 1)**
>
> We appreciate your approval of our idea and the detailed and insightful comments. Your concerns will be addressed in the following comments and the final version of our paper will be updated accordingly.
>
> 📝  **Q:  Evaluation with pre-existing evaluation protocols.**
>
> 💡&#8194;**A:**  Following the reviewer's suggestion, we conduct new experiments following existing evaluation methods like VolSDF. As these methods only provide PSNR metric results in their papers, we thus also compare with them in terms of PSNR but meanwhile report the SSIM and LPIPS of our approach. As shown in table below, our method can also achieve good results on pre-existing evaluation protocols.
>
> | |NeRF |VolSDF |Ours| Ours |Ours|
> |:--:|:--:|:--:|:--:|:--:|:--:|
> |Scan |PSNR↑ |PSNR↑ |PSNR↑ |SSIM↑| LPIPS↓|
> |24 |26.24| 26.28| **27.42** |0.813 |0.207|
> |37 |25.74| 25.61 |**26.32** |0.692 |0.223|
> |40 |26.79 |26.55 |**26.86** |0.569 |0.225|
> |55 |**27.57** |26.76 |26.98 |0.854 |0.307|
> |63 |31.96 |31.57 |**31.99** |0.803 |0.199|
> |65 |31.50 |31.50 |**32.63** |0.879 |0.268|
> |69 |29.58 |29.38 |**30.27** |0.939 |0.273|
> |83 |32.78 |33.23 |**34.34** |0.941 |0.333|
> |97 |28.35 |28.03 |**28.96** |0.913 |0.313|
> |105 |32.08| 32.13| **33.01** |0.932| 0.333|
> |106 |33.49 |33.16 |**33.77** |0.945 |0.332|
> |110 |31.54 |31.49 |**32.01** |0.943 |0.383|
> |114 |31.00 |30.33| **31.18** |0.916 |0.321|
> |118 |35.59 |34.90 |**35.63** |0.956 |0.322|
> |122 |35.51 |34.75 |**35.66** |0.960 |0.337|
> |Mean| 30.65 |30.38 |**31.14** |0.870 |0.292|
>
> 📝  **Q:  Fig.1. in the supplementary material very helpful to understand the proposed architecture.**
>
> 💡&#8194;**A:**  Thanks for your suggestions. We will remove the repeated part compared with Fig.1. (in the paper), then add Fig.1. (in the supplementary material) to the final version for a better understanding.
>
> 📝  **Q:  What is the intuition behind using the VQGAN pre-trained codebook?  Would it be better to use a pre-trained codebook from a 3D network?**
>
> 💡&#8194;**A:**  The intuition behind the VQ-quantized codebook prior is introducing additional high-level semantic descriptors of the context scene as the prior information, to enrich the feature representation of each coordinate and compensate for the missing views when only sparse views are available. In our experiments, we borrow the codebook from the 2D-aware dataset without introducing additional 3D prior for a more fair comparison. To verify its effectiveness, we replace the pre-training codebook of the NIR network with random initialization under both sparse views and few views settings and report the results in table below. We can see that the performance drops significantly without introducing 2D priors. And the robustness is reduced substantially with fewer views because it fails on scan24 under the few views setting.
>
> [1] Hou J, Xie S, Graham B, et al. Pri3d: Can 3d priors help 2d representation learning?[C]//Proceedings of the IEEE/CVF International Conference on Computer Vision. 2021: 5693-5702.
>
> [2] Liu Y, Wang L, Liu M. Yolostereo3d: A step back to 2d for efficient stereo 3d detection[C]//2021 IEEE International Conference on Robotics and Automation (ICRA). IEEE, 2021: 13018-13024.
>
> To verify this, we replace the pre-training codebook of the Implicit Neural Representation network with random initialization under both sparse views and few views settings.
>
> |Method with few views (5-8) |PSNR↑  |SSIM↑  |LPIPS↓ |
> |:--:|:--:|:--:|:--:|
>  |Without INR codebook  |13.979  |0.564  |0.298 |
>  |Randomly initialized INR codebook  |11.941  |0.512  |0.327 |
>  |**Ours (pre-trained codebook)**  |**15.622**  |**0.576**  |**0.283** |
>
>
>   |Method with sparse views (16-32)   |PSNR↑   |SSIM↑   |LPIPS↓  |
> |:--:|:--:|:--:|:--:|
>   |Randomly initialized INR codebook  | 22.307   |0.615  | **0.229**  |
>   |**Ours (pre-trained codebook)**  |**22.907**   |**0.688**   |**0.229**  |
>
> Besides, we have noticed that recent works like 3D-RETR[1], AutoSDF [2], and ShapeFormer[3], have successfully trained the VQ-quantized codebook from a 3D-aware dataset like ShapeNet. We believe that introducing 3D-aware codebook priors could further boost the performance, especially for geometry. But the exploring on how to train a 3D-aware codebook prior and leverage it to serve NERF task is out of this work's focus.
>
> [1] 3D-RETR: End-to-End Single and Multi-View 3D Reconstruction with Transformers, Zai Shi, Zhao Meng, Yiran Xing, Yunpu Ma, and Roger Wattenhofer. BMVC 2021.
>
> [2] AutoSDF: Shape Priors for 3D Completion, Reconstruction and Generation, Paritosh Mittal, YenChi Cheng, Maneesh Singh, and Shubham Tulsiani. CVPR 2022.
>
> [3] ShapeFormer: Transformer-based Shape Completion via Sparse Representation, Xingguang Yan and Liqiang Lin and Niloy J. Mitra and Dani Lischinski and Danny Cohen-Or and Hui Huang. Arxiv.2201.10326.

---

### Official Review · Reviewer_q1CY · 2022-07-09

**Rating:** 5
**Confidence:** 4
**Soundness:** 2 fair
**Presentation:** 2 fair
**Contribution:** 2 fair

**Summary:**

This paper focus on improving the training efficiency of coordinate based representations by reducing the number of camera views needed during training. To accomplish this, the authors proposed a codebook attention module and a coordinate attention module to inject prior knowledge into implicit representations. Experiments on novel view synthesis are conducted and improvements are shown over baseline methods on some public datasets.

**Questions:**

- In Table 1. the quantitative results from the presented method and the VolSDF are tied on DTU dataset. However, it seems like in fig 2. the qualitative results are having a larger gap there. I am wondering what is the cause for this?


**Limitations:**

- Although prior knowledge are used in this work, it seems like the model is still trained specifically to one scene and is hard to generalize to novel scenes.

**Strengths And Weaknesses:**

Strengths:

- The idea of bringing prior knowledge into implicit representation is novel and interesting.

- The paper is well written and easy to read.

Weaknesses:

- Although we do observe improvements over other baselines, it seems like the changes are relatively small. For example, in fig.2 the results of VolSDF and the presented methods are giving similar rendering results and in table.1 the quantitative results are tied.

- Lack of some ablation studies for a more thorough analysis of the proposed method. For example, the presented methods added two new modules to baseline implicit representations. And when doing comparisons with baseline methods, it is hard to tell whether it is because the addition of the new module brings the improvement or it is probably the network has more weights and could fits better. It could be better if more details are provided, for instance, the number of weights for the presented method and the baselines.

---

> ### Author Response · Authors · 2022-08-02
> **Authors Response (Part 2)**
>
> 📝  **Q: More ablation studies**: Per-scene Learnable Codebook vs. Fixed Pre-trained Codebook vs. Without the codebook attention module.
>
> 💡&#8194;**A:** Experiment results of either using randomly initialized codebook or without codebook are shown in table below. The performance and qualitative quality of both settings degrade significantly, and some scans fail with few views. Therefore, our prior can provide stronger robustness and better rendered images, especially with fewer views.
>
>  |Method  with few views (5-8) |PSNR↑  |SSIM↑ | LPIPS↓ |
> |:--:|:--:|:--:|:--:|
>  |Without codebook  |12.694  |0.523  |0.325 |
>  |Randomly initialized codebook  |13.315 | 0.525  |0.298 |
>  |**Ours (pre-trained codebook)**  |**15.622**  |**0.576** | **0.283** |
>
> 📝  **Q: More ablation studies**: Reduce the number of MLP layers (even to 0).
>
> 💡&#8194;**A:**  Experiment results of reducing the number of MLP layers in the Implicit Neural Representations module are shown in table below. Benefiting from our CoCo-Attention module, which queries representative features from per-scene relevant learnable embeddings for each coordinate, new views can be synthesized even with 1 MLP layer under different settings of view sparsity. But using 0 MLP layers will not work because we need at least 1 MLP layer to model a scene as the foreground object into a bounded sphere like NeRF++.
>   |Method  with few views (5-8) | PSNR↑   |SSIM↑   |LPIPS↓  |
> |:--:|:--:|:--:|:--:|
>   |2 layers MLP   |14.656   |0.560  | 0.292  |
>   |1 layers MLP  | 13.210  | **0.576**   |0.311  |
>   |0 layers MLP   |7.309  | 0.303   |0.401  |
>   |**Ours (4 layers MLP)** |**15.622**  | **0.576**  | **0.283**  |
>
> 📝  **Q: More ablation studies**: The number of coordinates attention modules.
>
> 💡&#8194;**A:**  We perform the ablation studies with using more coordinate attention modules in the Implicit Neural Representation (geometry) and fewer in the Neural Renderer (color) networks in table below. We can see that directly using fewer modules in Neural Renderer would decrease the performance due to the reduced color representation. While directly using more modules in Implicit Neural Representation would also reduce the performance since the RGB supervision applied to the color part, instantly increasing the geometry network’s parameters might result in overfitting.
>   |Method with few views (5-8)  | PSNR↑   |SSIM↑   |LPIPS↓  |
> |:--:|:--:|:--:|:--:|
>   |INR #1, ↓NR #1 |14.840  | 0.560  | 0.291  |
>   |↑INR #2, NR #2  |14.187  | 0.553  | 0.287  |
>   |↑INR #2, ↓NR #1 |13.946  | 0.528  | 0.323  |
>   |**Ours(INR #1, NR #2)**  | **15.622**  | **0.576**  | **0.283**  |

---

> ### Author Response · Authors · 2022-08-02
> **Authors Response (Part 1)**
>
> We appreciate your approval of our idea and the detailed and insightful comments. Your concerns will be addressed in the following comments and the final version of our paper will be updated accordingly.
>
> 📝  **Q:  Compared to volsdf, quantitative results are tied , but qualitative results are having a larger gap.**
>
> 💡&#8194;**A:**  We would like to clarify the results issue from two aspects as follows.
>
> **1)**. Effectiveness of our method:
> Our method is more robust and preserves more precise details under the few and sparse input views than VolSDF. VolSDF has failed on some scans, such as DTU scan118 (sparse views), DTU 63 (few views), BlendedMVS scan6 (few views), etc. However, our method produces decent results with precise details among these scenes, as shown in Fig. 2. For quantitative analysis (see table below) on the BlendedMVS and H3DS datasets which have richer texture information, our method achieves consistently significant increase of PSNR with more than 3\%, and LPIPS with more than 5\%.
>
> | |DTU |  DTU  |DTU | BlendedMVS  | BlendedMVS |  BlendedMVS | H3DS | H3DS | H3DS |
> |:--:|:--:|:--:|:--:|:--:|:--:|:--:|:--:|:--:|:--:|
>  | |PSNR↑ | SSIM↑ | LPIPS↓ | PSNR↑  |SSIM↑  |LPIPS↓  |PSNR↑  |SSIM↑  |LPIPS↓ |
>  |VolSDF |26.609  |0.839 | 0.309 | 18.942 | 0.747 | 0.213 | 23.922 | 0.898 | 0.110 |
>  |**Ours**  |26.738  |0.852 | 0.298 | 19.594 | 0.764 | 0.201  |25.279 | 0.911 | 0.098 |
>  |**Improvement(%)** | **0.4** | **1.5**  |**3.5** | **3.4** | **2.3** | **5.6**  |**5.7** | **1.4** | **10.9** |
>
> **2)**.Evaluation Metrics.
>
> Visual similarity is very subjective and aims to mimic human visual perception. Simple metrics like PSNR and SSIM are insufficient to assess an image's perceptual quality [1,2]. A well-known example is that blurring causes significant perceptual artifact but with small $L2$ change. For our method, it can be seen that the performance of our method on DTU scan118 is much better than VolSDF as observed from Fig.2., but our method's PSNR and SSIM score is lower than that of VolSDF. So we introduce LPIPS as evaluation metrics in the paper. It can be seen that on each dataset, the LPIPS of our method is over 3\% higher than that of VolSDF, which indicates a more meaningful metric.
>
> [1] Zhang R, Isola P, Efros A A, et al. The unreasonable effectiveness of deep features as a perceptual metric[C]//Proceedings of the IEEE conference on computer vision and pattern recognition. 2018: 586-595.
>
> [2] Ma Y, Zhai Y, Yang C, et al. Variable Rate ROI Image Compression Optimized for Visual Quality[C]//Proceedings of the IEEE/CVF Conference on Computer Vision and Pattern Recognition. 2021: 1936-1940.
>
> 📝  **Q: Performance improvement due to large model size or design.**
>
> 💡&#8194;**A:**  To verify whether the improvement comes from our proposed CoCo modules or additional parameters, we double the dimension and number of MLP layers of VolSDF (called enlarge VolSDF), so that its parameter size (6.21M) is approximately the same as our method (6.16M). Table shows the experimental results. Directly stacking parameters can not get a significant performance improvement. In fact, due to the sparsity of training views, the network is easy to overfit, and the enlarged VolSDF often outputs misplaced images (images that are close to the verification view but actually belong to the training view). However, our method uses a large ratio of parameters to query learnable embeddings from the codebook. These parameters are only related to generating a scan-related prior and do not directly operate with coordinates, which can maintain the generalization of new perspectives and prevent over-fitting.
>
> |Method with few views (5-8)| PSNR↑| SSIM↑ |LPIPS↓|
> |:--:|:--:|:--:|:--:|
> |VolSDF| 14.249 |0.557| 0.290|
> |Enlarge VolSDF |14.322| 0.563| 0.294|
> |**Ours** |**15.622** |**0.576** |**0.283**|
>
> 📝  **Q:  The model is hard to generalize to novel scenes.**
>
> 💡&#8194;**A:** In this work, we focus on how to bring additional prior information for implicit neural representation networks. We have successfully demonstrated the effectiveness of the dataset-related priors from the VQ-quantized codebook in ImageNet for the scene reconstruction and novel view synthesis tasks. However, the cross-scene generalization of INR is another critical area that needs more attention in the future. Our method has the potential to be extended to cross-scene generalization by introducing the scene-related codebook priors, pre-training on a cross-scene dataset, and finetuning with faster convergence on a novel scene, like Pixel-NeRF.

---

### Official Review · Reviewer_tT43 · 2022-07-11

**Rating:** 5
**Confidence:** 3
**Soundness:** 2 fair
**Presentation:** 2 fair
**Contribution:** 3 good

**Summary:**

The paper presents a method to utilize a non-scene-specific ImageNet-pretrained codebook for learning neural 3D representations, such as NeRFs. The codebook attention module transforms the codebook into a scene descriptor, and the coordinate attention module concatenates it with the positional encoding/features. The intuition is that doing so encourages the network to learn the semantic correlation between the input point (e.g. R^3) and the scene, enabling "extrapolation".

**Questions:**

How did you get the baseline MLP features (e.g. compared in Figure 4)?

**Limitations:**

The paper does not address limitations and societal impact.

**Strengths And Weaknesses:**

Strength: Demonstrates that the codebook method from VQGAN and VQ-VAE can be applied to 3D representation learning, lifting 2D ImageNet-based features to 3D. While the architecture may seem incremental, the extension to 3D is noteworthy.


Weakness: No quantitative ablation study. The "per-scene learnable features vs. fixed codebook" and "impact of codebook attention" experiments are both important yet entirely qualitative. Improvement from VolSDF could have been due to better initialization, more computation, and so on (as usual).  It would also be helpful to empirically evaluate novel views vs. observed views.

---

> ### Author Response · Authors · 2022-08-02
> **Authors Response (Part 2)**
>
> 📝  **Q: More ablation studies**: Reduce the number of MLP layers (even to 0).
>
> 💡&#8194;**A:**  Experiment results of reducing the number of MLP layers in the Implicit Neural Representations module are shown in table below. Benefiting from our CoCo-Attention module, which queries representative features from per-scene relevant learnable embeddings for each coordinate, new views can be synthesized even with 1 MLP layer under different settings of view sparsity. But using 0 MLP layers will not work because we need at least 1 MLP layer to model a scene as the foreground object into a bounded sphere like NeRF++.
>   |Method  with few views (5-8) | PSNR↑   |SSIM↑   |LPIPS↓  |
> |:--:|:--:|:--:|:--:|
>   |2 layers MLP   |14.656   |0.560  | 0.292  |
>   |1 layers MLP  | 13.210  | **0.576**   |0.311  |
>   |0 layers MLP   |7.309  | 0.303   |0.401  |
>   |**Ours (4 layers MLP)** |**15.622**  | **0.576**  | **0.283**  |
>
> 📝  **Q: More ablation studies**: The number of coordinates attention modules.
>
> 💡&#8194;**A:**  We perform the ablation studies with using more coordinate attention modules in the Implicit Neural Representation (geometry) and fewer in the Neural Renderer (color) networks in table below. We can see that directly using fewer modules in Neural Renderer would decrease the performance due to the reduced color representation. While directly using more modules in Implicit Neural Representation would also reduce the performance since the RGB supervision applied to the color part, instantly increasing the geometry network’s parameters might result in overfitting.
>   |Method with few views (5-8)  | PSNR↑   |SSIM↑   |LPIPS↓  |
> |:--:|:--:|:--:|:--:|
>   |INR #1, ↓NR #1 |14.840  | 0.560  | 0.291  |
>   |↑INR #2, NR #2  |14.187  | 0.553  | 0.287  |
>   |↑INR #2, ↓NR #1 |13.946  | 0.528  | 0.323  |
>   |**Ours(INR #1, NR #2)**  | **15.622**  | **0.576**  | **0.283**  |
>
> 📝  **Q:  Limitations and societal impact**
>
> 💡&#8194;**A:**  We have discussed the limitations in the Supplementary Materials and will merge them into the final version. Our method has conducted human face-related experiments and generates high-fidelity 3D-aware images from sparse images. Since human faces are highly private, our work might have some negative societal impacts for malicious purposes. We will add this societal impact into the final version.

---

> ### Author Response · Authors · 2022-08-02
> **Authors Response (Part 1)**
>
> We appreciate your approval of our idea and the detailed and insightful comments. Your concerns will be addressed in the following comments and the final version of our paper will be updated accordingly.
>
> 📝  **Q: How did you get the baseline MLP features**
>
> 💡&#8194;**A:**  To demonstrate the feature representation of each coordinate, we firstly sample two local patches with 16 x 16 pixels of two foreground objects (one is a yellow apple, and the other is a red one) from DTU 63. Then, for each pixel ray in the local patches, we sample 128 points along each ray. We separately feed these sampled ray points into our CoCo-VolSDF and pure MLP-based network (MLP-VolSDF). Next, we extract the coordinate features with dim-256 (before the last regressor layer to SDF/density or color) in the geometry network (color network) of both CoCo-VolSDF and pure MLP-VolSDF. Then, we project these coordinate features into the visual 3D space via t-SNE and finally arrive at the visualizations in Fig. 4 of the paper. Our proposed CoCo-based network results in more discriminative features for each coordinate. It indicates that the codebook prior could enrich each coordinate's feature representation.
>
> 📝  **Q: Performance improvement due to large model size or design.**
>
> 💡&#8194;**A:**  To verify whether the improvement comes from our proposed CoCo modules or additional parameters, we double the dimension and number of MLP layers of VolSDF (called enlarge VolSDF), so that its parameter size (6.21M) is approximately the same as our method (6.16M). Table below shows the experimental results. Directly stacking parameters can not get a significant performance improvement. In fact, due to the sparsity of training views, the network is easy to overfit, and the enlarged VolSDF often outputs misplaced images (images that are close to the verification view but actually belong to the training view). However, our method uses a large ratio of parameters to query learnable embeddings from the codebook. These parameters are only related to generating a scan-related prior and do not directly operate with coordinates, which can maintain the generalization of new perspectives and prevent over-fitting.
>
> |Method with few views (5-8)| PSNR↑| SSIM↑ |LPIPS↓|
> |:--:|:--:|:--:|:--:|
> |VolSDF| 14.249 |0.557| 0.290|
> |Enlarge VolSDF |14.322| 0.563| 0.294|
> |Ours |**15.622** |**0.576** |**0.283**|
>
> 📝  **Q: More ablation studies**: Per-scene Learnable Codebook vs. Fixed Pre-trained Codebook vs. Without the codebook attention module.
>
> 💡&#8194;**A:** Experiment results of either using randomly initialized codebook or without codebook are shown in table below. The performance and qualitative quality of both settings degrade significantly, and some scans fail with few views. Therefore, our prior can provide stronger robustness and better rendered images, especially with fewer views.
>
>  |Method  with few views (5-8) |PSNR↑  |SSIM↑ | LPIPS↓ |
> |:--:|:--:|:--:|:--:|
>  |Without codebook  |12.694  |0.523  |0.325 |
>  |Randomly initialized codebook  |13.315 | 0.525  |0.298 |
>  |**Ours (pre-trained codebook)**  |**15.622**  |**0.576** | **0.283** |

---

### Official Review · Reviewer_Jhwv · 2022-07-13

**Rating:** 7
**Confidence:** 4
**Soundness:** 4 excellent
**Presentation:** 3 good
**Contribution:** 3 good

**Summary:**

This paper proposed CoCo-INR, an implciit neural representation trained with sparse multi-view images leveraging on prior information from pre-trained image prototypes/features.

Specifically, two attention modules called codebook attention and coordinate attention are proposed to introduce most revelant scene priors into the learning process of neural implicit representations, so that few images are adequate to render realistic images as well as high-quality geometric information. Three datasets of different complixities are used to demonstrate the effectiveness of method.

**Questions:**

1. One important missing experiments are about other strategies in introducing scene priors like PixelNeRF and GRF. Existing experiments only consider baseline methods without external piros so the comparison is not that fair in my point of view.


2. Further demonstrate on how CoCO-VolSDF and CoCo-NeRF separately represent the full scene and individual improvements among baseline methods are expected.

3. More ablation on hyper-parameters are welcome.

4. Please consider adding more details about the training set-ups, e.g., how the subset views are selected. Furthermore, depending on how the training subset are selected, it would be good to show if it is possible to infer images given extrapolated view points.


**Limitations:**


Some limitations are mentioned in the supplement material. Related to that, does the method scale to larger or unbounded scenes well? Is the cluterred background of these scenes hamper the performance of CoCo-INR? It woukd be good to add some analysis here.

**Strengths And Weaknesses:**

Strengths

1. The idea of integrating VQ codebook into NeRF training looks interesting and leads to promising geometry recontruction given few images on three public datasets.

2. The experimental set-up of ablations and vislusations validate the effectiveness of codebook attention.

3. Though there are some missing details (mentioned in the weakness part), the overall writing is good and easy to read.

Weaknesses


1. How to select a subset of training views are not clearly represented and only the number of views are listed. Are subset views evenly sampled are consecutively sampled while leaving the rest as testing views?

    If possible, it would be good to see the camera trajectory or distributions along with rendering results to get better understand the method.

2. Similar to previous points, as cross-attention mechanism are adopted to propagate information from observed to unobserved ones, it would be interesting to see the extrapolated view synthesis in addition to interpolation.

3. One major point missing in my opinion is the comparison to other baselines introducing learnable/pre-trained priors into training process, such as PixelNeRF (CVPR2021) and GRF (ICCV2021).

    How does the codebook prior using VQ compared to stratigies adopted in PixelNeRF or GRF, where CNN feautes are used to augment local priors so that few views could be used to enable view synthesis. I think it is necessary to add related comparisons and discussions since currently all selected baselines does not take any external priors.

4. Another interesting thing is to explore is how the richness of priors affect performance of CoCo-INR? For example,the number of embeddings M and the size or Codebook, for a specific given scene, presumably a limited number of prototypes would be activated while others are not. Will a meaningful subset reduce the overall computation without obvious performance degradation?

5. As Coco-VolSDF and CoCo-NeRF are used to model foreground and background respectively, following NeRF++. It would good to see how the decomposition looks like because it is not clear to readers how the CoCo-NeRF hanles background regions. In Figure 2 and 3, there are some noisy patterns in the first two images while the rest shows a clean backgroud geometry.Is the geometry (normal) only computed from foreground regions? Is the grey region computed from background NeRF? How does the attention mechism helps the background region?


6. Minor Issues.
In Sec3.2 of supplement, is it claimed that only 3 extreme views are used in Fig.2 while the captions says 8 images are used. Please clarify how many images are used?

---

> ### Author Response · Authors · 2022-08-02
> **Authors Response (Part 3)**
>
>
> 📝  **Q:How the richness of priors affect performance of CoCo-INR.**
>
> 💡&#8194;**A:** We report the results of different numbers of codebook items, ranging from 16384, 8192, 4096, 2048 to 1024 in table below. It shows that the performance tends to decrease with the reduction of the codebook items.
> |Method  with few views (5-8)  |PSNR↑  | SSIM↑  | LPIPS↓  |
> |:--:|:--:|:--:|:--:|
>   |8192 items  | 15.463  | 0.567   |0.287  |
>   |4096 items  | 14.042  | 0.550   |0.288  |
>   |2048 items  | 14.762   |0.565   |0.285  |
>   |1024 items  | 13.948  | 0.548   |0.300  |
>   |**Ours(16384 items)**   |**15.622**  | **0.576**   |**0.283**|
>
> 📝  **Q: More ablation studies**: Per-scene Learnable Codebook vs. Fixed Pre-trained Codebook vs. Without the codebook attention module.
>
> 💡&#8194;**A:** Experiment results of either using randomly initialized codebook or without codebook are shown in table below. The performance and qualitative quality of both settings degrade significantly, and some scans fail with few views. Therefore, our prior can provide stronger robustness and better rendered images, especially with fewer views.
>
>  |Method  with few views (5-8) |PSNR↑  |SSIM↑ | LPIPS↓ |
> |:--:|:--:|:--:|:--:|
>  |Without codebook  |12.694  |0.523  |0.325 |
>  |Randomly initialized codebook  |13.315 | 0.525  |0.298 |
>  |**Ours (pre-trained codebook)**  |**15.622**  |**0.576** | **0.283** |
>
> 📝  **Q: More ablation studies**: Reduce the number of MLP layers (even to 0).
>
> 💡&#8194;**A:**  Experiment results of reducing the number of MLP layers in the Implicit Neural Representations module are shown in table below. Benefiting from our CoCo-Attention module, which queries representative features from per-scene relevant learnable embeddings for each coordinate, new views can be synthesized even with 1 MLP layer under different settings of view sparsity. But using 0 MLP layers will not work because we need at least 1 MLP layer to model a scene as the foreground object into a bounded sphere like NeRF++.
>   |Method  with few views (5-8) | PSNR↑   |SSIM↑   |LPIPS↓  |
> |:--:|:--:|:--:|:--:|
>   |2 layers MLP   |14.656   |0.560  | 0.292  |
>   |1 layers MLP  | 13.210  | **0.576**   |0.311  |
>   |0 layers MLP   |7.309  | 0.303   |0.401  |
>   |**Ours (4 layers MLP)** |**15.622**  | **0.576**  | **0.283**  |
>
> 📝  **Q: More ablation studies**: The number of coordinates attention modules.
>
> 💡&#8194;**A:**  We perform the ablation studies with using more coordinate attention modules in the Implicit Neural Representation (geometry) and fewer in the Neural Renderer (color) networks in table below. We can see that directly using fewer modules in Neural Renderer would decrease the performance due to the reduced color representation. While directly using more modules in Implicit Neural Representation would also reduce the performance since the RGB supervision applied to the color part, instantly increasing the geometry network’s parameters might result in overfitting.
>   |Method with few views (5-8)  | PSNR↑   |SSIM↑   |LPIPS↓  |
> |:--:|:--:|:--:|:--:|
>   |INR #1, ↓NR #1 |14.840  | 0.560  | 0.291  |
>   |↑INR #2, NR #2  |14.187  | 0.553  | 0.287  |
>   |↑INR #2, ↓NR #1 |13.946  | 0.528  | 0.323  |
>   |**Ours(INR #1, NR #2)**  | **15.622**  | **0.576**  | **0.283**  |
>
>
>
>
> 📝  **Q: Figure 2 in the Supplementary Material has an incorrect title.**
>
> 💡&#8194;**A:**  Thanks for pointing out this typo. It should be "Qualitative visualization results (zoom-in for the best of views) on the DTU dataset with 3 extremely limited views." We will correct it in the final version.
>
>
> 📝  **Q: Does the method scale to larger or unbounded scenes well?**
>
> 💡&#8194;**A:**  Thanks. Our proposed CoCo-INR can be applied to large-scale or unbounded scenes following the NeRF++'s foreground and background parameterization. The experiment results on the BlendMVS dataset (scenes with foreground objects and unbounded background) have demonstrated our method's capability to model unbounded scenes.
>
> &#8194;&#8194;&#8194;&#8194;For a large-scale scene such as a city, our CoCo-INR could introduce different local codebook priors from each street block to separately model the local regions of a city. With the help of local codebook priors and the powerful representation ability of the transformer-based Multi-Head Attention mechanism, our CoCo-INR will be able to model a city-level scene. We believe it could be a promising research direction and will inspire more solutions in the future.

---

> > ### Comment · Reviewer_Jhwv · 2022-08-08
> > **Reviewer Reponse After Rebuttal**
> >
> > Thank authors for the detailed response to my questions and concerns.
> >
> > I appreciate the clarification of Coco-INR compared against approaches relying on scene-related priors. Additional evaluation and abalation of key components also adressed most of my concerns.
> >
> > After reading all the reviews and authors' feedbacks, I would like to keep my positive rate towards this paper and  recommend an acceptance.

---

> > > ### Author Response · Authors · 2022-08-08
> > > **Authors Response After Rebuttal**
> > >
> > > We appreciate your valuable and insightful comments. We feel glad about your generally favorable assessment of our methodology. Additional evaluation/ablation and corresponding explanations will be included in the final version.

---

> ### Author Response · Authors · 2022-08-02
> **Authors Response (Part 2)**
>
>
> 📝  **Q:  Comparison to other baselines introducing learnable/pre-trained priors.**
>
> 💡&#8194;**A:** We have also noticed these works about NeRF generalization. Even though these works can reconstruct new scenes/objects based on a small number of views and show certain generalizations, the priors they use differ from ours. Specifically, the priors in these works are learned from many scans in the same dataset with the testing scenes, and the performance thus heavily depends on whether priors and testing scenes have a high semantic and appearance correlation. But our prior is a codebook obtained by training VQ-GAN on the 2D-aware dataset, ImageNet, which is not specially designed for a particular scene/object or even a specific 3D dataset, thus relieving the NeRF model of over-dependence on the training-testing scene consistency. Nevertheless, we follow the review comment and conduct new experiments as follows.
>
>  &#8195; &#8195;In Pixel-NeRF's experimental protocols, many scans share the same object in the DTU dataset. We take the first test scene (scan 8) of Pixel-NeRF on the DTU dataset as an example for experiments. Scan 8 and scan 51 are two scans sharing the same object, but scan 51 is for the training, and scan 8 is for testing in Pixel-NeRF. These priors in Pixel-NeRF share a high-correlation context (the same object) between the training and testing scenes, while our codebook priors are scene agnostic. Our CoCo-INR can also learn the scene-related priors, same with Pixel-NeRF, and we perform the experiments of our method with scene-related priors and scene-agnostic priors as shown in table below. It shows that our CoCo-INR with scene-related priors outperforms the Pixel-NeRF under the same setting. Our CoCo-INR with scene-agnostic priors achieves a relatively comparable performance with Pixel-NeRF with scene-related priors.
>
> |Method |PSNR↑ |SSIM↑ |LPIPS↓|
> |:-:|:-:|:-:|:-:|
> Pixel-NeRF (with scan 51 priors) |19.927| **0.776** |0.202|
> **Ours (without scan 51 priors)** |18.263 |0.673 |0.173|
> **Ours (with scan 51 priors)**| **29.583** |0.708 |**0.125**|
>
>  &#8195; &#8195;Meanwhile, we further design a relatively fair comparison experiment without scene intersection between training and testing. We use the Pixel-NeRF pre-trained model on the DTU dataset and test on the BlendedMVS dataset. Pixel-NeRF introduces multi-view (3D) priors from the DTU dataset, but our method only introduces 2D priors from the ImageNet dataset and tests on the BlendedMVS dataset.
>
>  | |Pixel Nerf| Ours| Pixel Nerf| Ours |Pixel Nerf |Ours|
> |:--:|:--:|:--:|:--:|:--:|:--:|:--:|
>  |Scan |PSNR↑| PSNR↑| SSIM↑ |SSIM↑ |LPIPS↓ |LPIPS↓|
>  |1 |9.329 | **16.597** | 0.329 | **0.630**  |0.377 | **0.265** |
>  |2  |9.188 | **15.277** | 0.318  |**0.637**  |0.338  |**0.303** |
>  |3  |10.180 | **10.965** | 0.303 | **0.490**  |0.473  |**0.398** |
>  |4 | 7.199 | **14.614**  |0.366 | **0.735** | 0.284  |**0.254** |
>  |5 | 8.308 | **14.358** | 0.352 | **0.699** | 0.355  |**0.268** |
>  |6 | 9.310 | **12.114** | 0.318  |**0.575** | **0.353** |0.366 |
>  |7 | 8.212 | **15.955** | 0.301  |**0.659** | 0.409  |**0.269** |
>  |8 | 12.831 | **16.384** | 0.505  |**0.759** | 0.347 | **0.178** |
>  |9 | 9.542  |**13.491** | 0.243 | **0.487** | 0.480 | **0.386** |
>  |Mean  |9.344 | **14.417** | 0.337 | **0.630**  |0.380 | **0.299** |
>
>  &#8195; &#8195;Table above shows that Pixel-NeRF does not perform as well as our method when both use cross-dataset priors. So even introducing priors from other datasets still cannot guarantee synthesizing novel views well. The design of prior learning and embedding, as well as equipping coordinates with richer features, plays a crucial role in NERF view synthesis, which is also our work's focus.
>
>
>
> 📝  **Q: How do CoCO-VolSDF and CoCo-NeRF separately represent the full scene?**
>
> 💡&#8194;**A:**  We follow NeRF++ to model a scene as the foreground object in a bounded sphere, and the unbounded background via an inverted sphere parameterization. The only difference between CoCo-VolSDF (foreground) and CoCo-NeRF is the type of geometry function (implicit network). We use the Signed Distance Function (SDF), the same as VolSDF, to model the foreground surface in CoCo-VolSDF, while we use the density function to model the unbounded background in CoCo-NeRF.
>
>  &#8195; &#8195;The grey regions in Fig.2 and Fig.3 (in the paper) with the "Normal" caption are empty regions predicted by the foreground network (CoCo-VolSDF). In contrast, these empty regions with colors in Fig.2 and Fig.3 with the "Render" caption are the predicted colors by the background network (CoCo-NeRF). The noisy patterns in the first two columns of the two figures are mainly due to the sparsely sampled training views of the DTU dataset with an ambiguity between foreground and background. However, since the sparsely sampled training views in BlendMVS and H3DS almost cover 360 degrees with a more apparent boundary between foreground and background, the noisy patterns disappear.

---

> ### Author Response · Authors · 2022-08-02
> **Authors Response (Part 1)**
>
> We appreciate your approval of our idea and the detailed and insightful comments. Your concerns will be addressed in the following comments and the final version of our paper will be updated accordingly.
>
> 📝 **Q: How to select training views and testing views.**
>
> 💡&#8194;**A:** We would like to divide view number into 3 settings: Sparse views (16-32 in total), Few views (5-8 in total) and Extremely limited views (only 3), and their sampling ways are as follows.
>
> For the first case, views are evenly sampled. In particular, for original scans with less than 64 views, we sample one from every two views for training according to the view ID, and the rest are used as testing views (i.e., for training: 0, 2, 4, 6...; for testing: 1, 3, 5, 7...). For original scans with more than 64 views of an object, the sampling interval is changed to $\lfloor \frac{\\#Views}{32} \rfloor$.
>
> For the second case, views are evenly sampled. In particular, for original scans with less than 64 views, we sample one from every eight for training according to the view ID, and the rest are used as testing views (i.e., for training: 0, 8, 16, 24...). For scans with more than 64 views, the sampling interval is changed to $\lfloor \frac{\\#Views}{8} \rfloor$.
>
> For the third case, we select three representative views for each scan as training views (usually left, right, and top views), and the remaining views are used for testing.
>
> As the comment box in the open review system does not support image and graphics input, we will provide the selected views lists and camera trajectory rendering images along with the code in the final version.
>
> 📝  **Q: New views are interpolation or extrapolation.**
>
> 💡&#8194;**A:** We would like to clarify that in the sparse views (16-32) setting, most testing views are interpolations. On the other hand, in the few and extremely limited views(3-8) settings, most testing views are extrapolations.

---

### Meta-Review · Area_Chair_eRmj · 2022-08-27

**Recommendation:** Accept
**Confidence:** Less certain

**Metareview:**

This paper focuses on improving the training efficiency of coordinate based representations by reducing the number of camera views needed during training. To accomplish this, the authors proposed a codebook attention module and a coordinate attention module to inject prior knowledge into implicit representations. The intuition is that doing so encourages the network to learn the semantic correlation between the input point and the scene, enabling "extrapolation" to far away views.

The reviewers appreciated the idea of the paper and how it improved image reconstruction quality across various number of views.  They raised concerns regarding lack of experiments with models that perform conditioning to pixel features, e.g., PixelNerf, to generalize across scenes, as well as lack of ablative quantitative experiments to evaluate the architectural contributions and lack of a limitations section. The rebuttal submitted by the authors includes experiments with PixelNerf but does not state mention the number of views used, and does not describe how the method “ours (with scan 51 priors)” is obtained.  The second experiment where PixelNerf is trained on DTU  and tested on BlendedMVS is not a fair experiment and we do expect pixelNerf to fail there given there is no test time adaptation through gradient descent at the test scene as is the case with the present method. The authors are encouraged to move all rebuttal experiments to the main paper, and thoroughly explain the experimental setup they used.
Overall, the paper is not very clearly written. Specifically, the reader  learns only at the end of the implementation detail section that a separate network is trained per scene.  Reviewer q1CY mentions: “Although prior knowledge are used in this work, *it seems like* the model is still trained specifically to one scene and is hard to generalize to novel scenes.”
By not comparing or contrasting the proposed approach to cross-scene generalization works the reader is left to wonder what is the generalization capabilities of the proposed model with varying number of input views.
The authors are encouraged to clarify these points in their final version.


**Award:**

No

---

### Decision · Program_Chairs · 2022-09-14

Accept